# Mathematical Modelling of Glyphosate Molecularly Imprinted Polymer-Based Microsensor with Multiple Phenomena

**DOI:** 10.3390/molecules27020493

**Published:** 2022-01-13

**Authors:** Fares Zouaoui, Saliha Bourouina-Bacha, Mustapha Bourouina, Nadia Zine, Abdelhamid Errachid, Nicole Jaffrezic-Renault

**Affiliations:** 1Institute of Analytical Sciences, University of Lyon, 69100 Villeurbanne, France; fareszou06@gmail.com (F.Z.); nadia.zine@univ-lyon1.fr (N.Z.); abdelhamid.errachid-el-salhi@univ-lyon1.fr (A.E.); 2Faculty of Technology, University of Bejaia, 06000 Bejaia, Algeria; lgebej@yahoo.fr (S.B.-B.); bouryas@yahoo.fr (M.B.)

**Keywords:** mathematical modelling, impedimetric microsensor, molecularly imprinted polymer, glyphosate

## Abstract

The massive and careless use of glyphosate (GLY) in agricultural production raises many questions regarding environmental pollution and health risks, it is then important to develop simple methods to detect it. Electrochemical impedance spectroscopy (EIS) is an effective analytical tool for characterizing properties at the electrode/electrolyte interface. It is useful as an analytical procedure, but it can also help in the interpretation of the involved fundamental electrochemical and electronic processes. In this study, the impedance data obtained experimentally for a microsensor based on molecularly imprinted chitosan graft on 4-aminophenylacetic acid for the detection of glyphosate was analyzed using an exact mathematical model based on physical theories. The procedure for modeling experimental responses is well explained. The analysis of the observed impedance response leads to estimations of the microscopic parameters linked to the faradic and capacitive current. The interaction of glyphosate molecules with the imprinted sites of the CS-MIPs film is observed in the high frequency range. The relative variation of the charge transfer resistance is proportional to the log of the concentration of glyphosate. The capacitance decreases as the concentration of glyphosate increases, which is explained by the discharging of the charged imprinted sites when the glyphosate molecule interacts with the imprinted sites through electrostatic interactions. The phenomenon of adsorption of the ions in the CMA film is observed in the low frequency range, this phenomenon being balanced by the electrostatic interaction of glyphosate with the imprinted sites in the CS-MIPs film.

## 1. Introduction

Glyphosate (GLY) is a non-selective foliar systemic herbicide [1]. GLY has become the most widely used phytosanitary molecule, especially as a weedkiller in agriculture. It is further used in various non-agricultural contexts such as forestry and household maintenance [2]. Glyphosate can cause harmful effects on animals [3,4], insects and birds [5], variety of animals in the aquatic system [6], microbial, fungal and bacterial biomasses [7]. The relationship between the use of GLY and adverse effects on human health has been the subject of numerous studies. In March 2015, the International Agency for Research on Cancer (IARC, Lyon, France), a department of the World Health Organization (WHO), classified glyphosate as “probably carcinogenic to humans”, in l ‘inserting into Category 2A, representing substances with limited evidence of carcinogenicity to humans and sufficient evidence to animals [8]. In May 2016, a joint expert meeting of WHO and the Food and Agriculture Organization of the United Nations (FAO) on pesticide residues in the environment and foods, concluded that “glyphosate is unlikely to present a carcinogenic risk to humans through the diet” [9].

Consequently, there is more interest in the development of selective, rapid, simple and inexpensive GLY detection methods. Electrochemical sensors are devices capable of meeting this demand. The main challenge for the electrochemical detection of glyphosate is its low electroactivity below an accessible potential window [10]. Therefore, in order to achieve low concentrations for the detection of GLY, the technique of immobilizing biological elements (antibody, enzyme, microorganism, DNA), on the surface of the working electrode, are generally used for GLY recognition to increase sensitivity and specificity, which are typically lacking in chemical sensors. However, as they are biomolecules, their use is limited to mild conditions of pH, temperature and nature of the medium in order to avoid denaturation of the protein. In addition, the use of these biological elements is not that simple. Their supply and immobilization at the surface of the electrodes are complex [11,12].

Due to the need to overcome these limitations, an alternative has been developed which consists of using molecularly imprinted polymers (MIPs). MIPs were used as a specific recognition material having binding sites of size and shape complementary to target molecules (template). To generate cavity-selective MIPs, a monomer is simply polymerized in the presence of the target molecule and a crosslinking agent resulting in the formation of a highly crosslinked template/monomer complex. MIPs are characterized by very interesting properties such as physical resistance, robustness, resistance to high pressures and temperatures and high inertia towards various chemicals. In addition, MIPs have high affinity and selectivity towards target molecules; they are comparable to natural receptors [13,14,15,16,17].

Sensors based on molecularly imprinted polymers (MIPs) are being developed for the detection of GLY. The functional monomers most often used in synthesis are MCA (N-methacryloyl-L-cysteine) [18] and PPy (Polypyrrole) [19,20].

In these works, the electrochemical detection of glyphosate was obtained by voltammetric methods. A more recent study, using MIP based on polyacrylamide cryogels, glyphosate was detected through capacitive measurements [21].

The choice of the functional monomer in the synthesis of MIPs is an essential step given its capacity to provide complementary interactions with the target molecules [22]. Chitosan (CS) is a natural polysaccharide obtained by deacetylation of chitin which is the most abundant non-toxic, biodegradable and biocompatible natural amino polysaccharide [23]. Chitosan has three types of reactive functional groups, an amine group and two primary and secondary hydroxyl groups. The presence of reactive functional groups on the polysaccharide chain of CS gives it flexibility, excellent film-forming capacity, good adhesion, biocompatibility, high mechanical resistance, possibility of undergoing structural modifications, which makes it suitable for the preparation of MIPs and for the construction of electrochemical sensors [24,25].

CS-MIPs electrochemical sensors have been reported in the state of the art. They are based on the electrodeposition of CS on the surface of the electrodes [26,27]. The electrodeposition of chitosan on a conductive surface by pH gradient causes release of H_2_. The blogging of these molecules inside the membrane makes it non-homogeneous [28]. The grafting of chitosan on the diazonium salt by the formation of strong interactions probably leads to more homogeneous membranes and to the resolution of interface problems [29].

Electrochemistry describes the chemical phenomena coupled with reciprocal exchanges of electrical energy when an electrolytic chemical medium interacts with an electrical circuit. It allows atomic scale transformations of matter and generations of different reactive species by transfer of electrons between an electrode and a substrate in an electrolytic solution controlled by an electric current or by a voltage [30]. Consequently, the exchange of electrons at the interface between an electronic conductor and the electrolytic medium containing oxidizer/reducer couples (Ox/Red) corresponds to the field of electrochemical reactions, while the reciprocal electron exchange reaction between different Ox/Red couples correspond to the field of oxidation-reduction reactions [31].

Electroactive species have several “modes of transport” to reach the site of the electron transfer reaction, namely the electrode / electrolyte interface such as convection, diffusion and migration. The phenomena related to the movement of matter can be approached at the microscopic level [30]. The electrochemical process can be affected by the phenomenon of adsorption on the surface of the electrode in contact with the solution. In terms of specific adsorption, electrosorption refers to the formation of a chemical bond between the charged species of the electrolyte and the surface of the electrode [32].

Several electrochemical methods are frequently used for physico-chemical analyzes of the electrode/electrolyte interface, such as conductimetry [33], potentiometry [34], amperometry [35], voltammetry [36], and impedimetric. These electrochemical techniques involve three essential quantities which are the current (I), the potential (E) and the time (t).

Electrochemical impedance spectroscopy (EIS) is one of the non-stationary techniques that allow the analysis of complex electrochemical systems. The EIS has established itself as a very powerful analytical tool given its great ability to study the interfacial properties of electrode/electrolyte [37]. The impedance method consists in imposing a sinusoidal disturbance in low amplitude potential, around a chosen voltage E_0_, between the working electrode and the reference electrode. The sinusoidal current response is thus measured with a continuous component I_0_, which corresponds to the potential E_0_ (I_0_ having a zero value when E_0_ is the dropout potential. This response is accompanied by a phase shift (φ) depending on the electrical elements present in the system. This system can be considered as a ≪black box≫, which reacts by emitting a signal I when it is subjected to a disturbance E [38].

This work was focused on the modeling of a glyphosate impedimetric microsensor. It was based on the covalent clawing of molecularly imprinted chitosan on 4-aminophenylacetic acid. Mathematical models based on physical theories are developed to analyze the data obtained experimentally. Analysis of the observed impedimetric response leads to the estimation of the microscopic parameters of the sensors. The validation of the different models is carried out by comparing the experimental and theoretical impedance. The response of ‘miniaturized developed’ sensors is closely related to the presence of specific molecular sites on the film of the biopolymer at the surface of the working macro-electrode, and also related to factors inherent in the structure of the sensor itself and in its mode of operation. Explaining the operation in the depths of this sensor requires in-depth knowledge of all the parameters involved in such a design.

Modeling is a simplified representation of a real physical system or phenomenon, making it possible to reproduce its functioning, to analyze, to explain and to predict certain aspects. Modeling is a tool which helps to understand the intrinsic mechanisms of these analytical instruments. It makes it possible to find relationships between the variables and the parameters that are supposed to influence the metrological characteristics of these sensors, for example the resistance of the solution, the resistance to charge transfer in the different ‘layers’ that make up the sensor, or geometry. Modeling is essential for optimization before a device or process goes to market. A “reliable” model is one that will simulate a sensor in real conditions. Therefore, there arises the problem of determining the parameters, which intervene in the equations of the model [39,40].

## 2. Materials and Methods

### 2.1. Reagents and Apparatus

Chitosan (CS, molecular weight 250 kDa with a degree of deacetylation 80–95%), ethanol (99.8%), methanol (purity 99.9%), hydrochloric acid (37%, HCl), Sulfuric acid (purity 95%), 1-Ethyl-3-(3-dimethylaminopropyl)carbodiimide (EDC), 4- aminophenylacetic acid (CMA), sodium nitrite (NaNO_2_), sodium hydroxide (NaOH) and glyphosate (GLY) were obtained from Sigma Aldrich, France. N-Hydroxysuccinimide (NHS) was purchased from Acros Organics, France. 

Multichannel potentiostat (Biologic-EC-Lab VMP3) analyser was used for all electrochemical measurements. Transducer with four bare-gold working microelectrodes (WE, S = 0.64 mm²/unit), two Ag/AgCl reference microelectrodes (RE, S = 0.13 mm²/unit) and one counter microelectrode (CE, S = 1.37 mm²). It was fabricated at the National Center for Microelectronics (CNM), CSIC, Barcelona, Spain. 

### 2.2. Preparation of the Microsensor

The Au microelectrodes were rinsed with ethanol and deionized water in ultrasound for 10 min then exposed to UV/ozone for 30 min. 

CMA solution (3 mM) was prepared in an aqueous solution of NaNO_2_ (20 mM) and HCl (20 mM). The mixture was stored at 4 °C for 5 min to slow down the chemical reaction between the different reagents. CMA was electrodeposited into gold microelectrodes by cyclic voltammetry for 5 cycles from −1.2 to 0.0 V at scan rate of 50 mV/s. After electrodeposition, the microelectrodes were rinsed with deionized water and dried using nitrogen. Carboxylic functions of CMA were activated by incubating of electrodes for 1 h in 1-ethyl-3- (3 dimethylaminopropyl) carbodiimide (EDC) (0.4 M) and N-Hydroxysuccinimide (NHS) (0.1 M) prepared in absolute ethanol. Then, the devices are rinsed carefully with pure ethanol and dried under light flow.

0.01 g/mL of CS was prepared in 0.1 M acetic acid and ultrasonicated until complete solubilization of CS. Then, 1 mg/mL of GLY was prepared in CS solution. This mixture was mixed for 2 h to obtain GLY/CS complex, then the pH of the solution was adjusted to 5 using 0.1 M NaOH. Drops of GLY/CS were deposited on the microelctrodes modified with CMA for 24 h. Microelectrodes were rinsed with deionized water then they were incubated in 0.5 M H_2_SO_4_ solution for 1 h to cross-link CS. In the end, GLY has been removed from CS by incubating microelectrodes in acetic acid/methanol solution (1:1, *v*/*v*) for 30 min. Thus, an electrochemical microsensor based on CS molecularly imprinted film modified with CMA was developed for the specific recognition of GLY.

The analytical results and characteristics of the developed microsensor are well presented and detailed in an earlier publication [29].

### 2.3. EIS detection Method

Microsensors was characterized using EIS (Initial potential E = 0.2 V. Highest Freq = 100 khz, Lowest Freq = 100 mHz) in phosphate buffer saline solution (PBS) with 5 mM ferro-ferricyanide ([Fe(CN)_6_]^3−/4−^) before and after incubation in deionized water containing different concentration of GLY (0.31 pg/mL to 50 ng/mL) for 30 min. EIS was used to investigate changes in charge transfer resistance of the film depending on the GLY concentration. 

## 3. Modeling of the Microsensor

### 3.1. Mathematical Model

In the models discussed below, the constituent phases have different conductivity values, but the conductivities are of the same type, they are all either ionic or electronic. The reaction at the electrode may include a charge transfer step with formation of ions adsorbed on the surface of the membranes. This is usually visualized by the presence of a three-phase boundary where the electronic conductor, ionic conductor, and pores meet [41].

The impedance spectrum characterizing the microsensor designed by grafting the molecularly imprinted chitosan onto the diazonium salt (CMA) and its equivalent circuit are shown in Figure 1.

The overall impedance spectrum forms two well separated semicircles: From 100 KHz to 17 Hz: This first semicircle is associated with a load transfer resistor (R_ct1_) in parallel with the constant phase elements (CPE1). At high frequency (100 KHz), the intersection of the impedance curve with the abscissa axis enables the electrolyte resistance *R*_s_ to be determined. From 17 Hz to 100 mHz: this second half-circle is modeled by a load transfer resistor (R_ct2_) in parallel with the constant phase elements (CPE2) [42]. 

The capacitive current is that associated with the charging or discharging of the electrochemical double layer. The capacitive current does not cause a change in the chemical composition, but only in the distribution of the interfacial electric charges. When a metal is placed in contact with an electrolyte at a given potential, an electrical interface is immediately developed. The area of the metal/solution interface where charged species exist is called the electrical double layer, which then behaves as if an electrical capacitor characterized by a capacitor C_dl_ [43]. Indeed, it has been established that surface heterogeneities of modified electrodes (roughness, porosity, presence of impurities, absorption on the double layer.) can be the cause of a frequency dispersion of the impedance spectra [44]. In these cases, Constant Phase Element (CPE) are used instead of capacitors to take into account the non-ideality of the system. In the literature, different definitions have been proposed. The CPE formula often used for adjustment is defined by the equation [45]:(1)$$Z_{\mathrm{CPE}}=\frac{1}{Q{\left(j\omega \right)}^{n}}$$
(2)$$j=\sqrt{-1}=\cos\left(\frac{\pi }{2}\right)+j\sin\left(\frac{\pi }{2}\right)$$

Q en (s^n^ Ω^−1^) ou (F s^(*n*−1)^) represents the CPE coefficient, −1 ≤ *n* ≤ 1 is the correction factor. ω = 2πf, where f represents the frequency measured in Hz.

In particular, Z_CPE_ represents an inductance if n = −1, a resistance if n = 0, a Warburg impedance if n = 0.5 and a capacitance if n = 1 [46,47]. Equation (1) can also be written:(3)$$Z_{\mathrm{CPE}}=\frac{1}{{Q\omega }^{n}}\left[\cos\left(n\frac{\pi }{2}\right)-j\sin\left(n\frac{\pi }{2}\right)\right]$$

The global real and imaginary impedances equivalent to the circuit of Figure 1 are given by the following equations:(4)Re(Z)=Rs+Rct1(1+Rct1Q1ωn1cosn1π2)1+(Rct1Q1ωn1)2+2Rct1Q1ωn1cosn1π2+Rct2(1+Rct2Q2ωn2cosn2π2)1+(Rct2Q2ωn2)2+2Rct2Q2ωn2cosn2π2
(5)−Im(Z)=Rct12Q1ωn1sinn1π21+(Rct1Q1ωn1)2+2Rct1Q1ωn1cosn1π2+Rct22Q2ωn2sinn2π21+(Rct2Q2ωn2)2+2Rct2Q2ωn2cosn2π2
where Re (Z) is the real part of the impedance, −Im (Z) is the imaginary part of the impedance. Q_1_ represents the coefficient of CPE1, n_1_ the correction factor of CPE1, Q_2_ is the coefficient of CPE2, n_2_ the correction factor of CPE2.

Changes in potential and current in the electrolyte lead to the concept of ohmic drop described as the electrolyte resistance R_s_. This resistance is an important factor contributing to the global impedance of the cell [48]. The impedance of the resistance of the solution is: (6)$$Z_{R_{S}}={\;R}_{s}$$
(7)$$R_{s}\left(Ω\right)=\;\rho \frac{l}{A}$$
where ρ resistivity of the solution (Ω cm), l the coating thickness and A is the area of the coating [49]. 

The resistivity of the ferri/ferrocyanide solution is calculated by the following relationships:(8)$$\rho =\frac{1}{\sigma }$$
(9)$$\sigma =\sum_{i}q_{i}\lambda_{i}C_{i}$$

With, σ: conductivity of the solution (s/m) who can be calculate from redox ion couple, C_i_ the concentration of the ion (mol/m^3^), q_i_ the number of charges of the ion, λ_i_ the equivalent molar ionic conductivity. 

The reactions at the electrodes involve the redox couple ferri/ferrocyanide, in which ferricyanide is the oxidant and the ferrocyanide is the reducing agent. For C_Ox_ = C_Red_ = C, and for a simple one-electron process (n = 1), R_ct_ is given by Equation (10):(10)$$R_{\mathrm{tc}}=\frac{\mathrm{RT}}{F^{2}{\mathrm{Ak}}^{0}C}$$

With; R: Ideal gas constant (J·mol^−1^·K^−1^), T: Temperature (K), F: Faraday constant (C·mo1^−1^), A: surface of the working electrode (cm²), k^0^ standard electron transfer rate constant (cm/s).

### 3.2. Numerical Simulation

To determine the various parameters and validate the proposed model, the response of the sensor was simulated by using Equations (1) and (2) using Matlab software. The parameters were varied with a very small step (Table 1), then the optimal values were determined by minimizing the error between the experimental values and the predicted values. The simulations should make it possible to find the link between the impedance measurement and the various parameters exposed. The developed program consists of two distinct parts. The first part consists in determining the parameters of the circuit R_s_ + CPE1/R_ct1_, after validation of this step the program proceeds to determine the parameters of the circuit CPE2/R_ct2_.

The error between the experimental and theoretical values is calculated according to the relations (11) and (12). The iterations are stopped when e_1_,e_2_ ≤ e_0_ (10^−3^).
(11)Error(e1)=∑|Re(Z)theoretical−Re(Z)experimentalRe(Z)theoretical|
(12)Error(e2)=∑|(−Im(Z))theoritical−(−Im(Z))experimental(−Im(Z))theoretical|

## 4. Results and Discussions

### 4.1. Model Validation

In order to validate the physical model, the simulation results are compared with the experimental data. We have obtained the results presented in Figure 2. We can first see that the results obtained theoretically are consistent with the experimental values obtained in ref [28]. Thus, the proposed physical model is in agreement with our simulations and it showed good agreement with the results obtained in the laboratory. Overall, the errors are less than 10^−3^.

### 4.2. Analysis of Theoretical Results

The values of each parameter of the model are presented in Table 2. The thickness of the membranes (CS + CMA) deposited on the electrode determined by the model is 36 µm. This value was found to vary of less than 1 µm from one sensor to another. This thickness is constant throughout the experiment indicating the stability of the sensor in the analysis medium. Thus, the resistance of the solution R_s_ is 151,14 Ohm, this resistance is also constant during the experiment.

For the parameters of the first semi-circle, the value of the standard rate constant k_1_° decreases with the increase in the concentration of the incubated GLY, going from 5,5 µm/s to 2 µm/s according to a linear law whose equation is (k _1_° = −0.284 log [GLY] +4.19) with a correlation coefficient R² = 0.997 (Figure 3a). The decrease in the electron transfer rate caused an increase in the charge transfer resistance R_ct1_, this linear variation is given by the following regression equation (k _1_° = −8.285 R_ct1_ + 5.35; R² = 0.961) (Figure 3b). The increase in R_ct1_ is proportional to the increase in GLY concentration. It went from 15132 Ohm for a zero GLY concentration to 41614 Ohm for a concentration of 50 ng/mL of incubated GLY (R_ct1_ = 3343.5 log [GLY] +14507.2; R² = 0.966) (Figure 3c). Moreover, the value of Q_1_, associated with the behavior of CPE1, is inversely proportional to the concentration of GLY going from 5.6 × 10^−7^ S^n^·Ω^−1^ ([GLY] = 0) to 4.56 × 10^−7^ S^n^·Ω^−1^ ([GLY] = 50 ng/mL) (Q_1_ = −0.16 × 10^−7^ log [GLY] + 5.83 × 10^−7^; R² = 0.977) (Figure 3d). The value of n_1_ which is 0.79 indicates that the surface is inhomogeneous. n_1_ being close to 1, it means that CPE1, included in the model, is nearly a pure capacitance.

The value of Q_1_ decreases with increasing GLY concentration. This decrease makes it possible to conclude that the capacitance of the CS-MIPs film and the concentration of GLY are inversely proportional. The same behavior has been observed in the state of the art where the capacitance of a MIPs sensor based on N, N-methylenebisacrylamide cryogel varies inversely proportional with the concentration of glyphosate [21]. The phenomenon was explained by the discharging of the charged imprinted sites when the glyphosate molecule interacts with the imprinted sites through electrostatic interactions.

Regarding the parameters of the second semi-circle, the rate constant k_2_° and the coefficient Q_2_ increase proportionally with the increase in the concentration of the incubated GLY. k_2_° went from 1.92 µm/s to 5.21 µm/s (k_2_° = 0.55 log [GLY] +0.78; R² = 0.986) (Figure 4a), while Q went from 1.07 × 10^−5^ S^n^·Ω^−1^ to 2.3 × 10^−5^ S^n^·Ω^−1^ (Q_2_ = 0.1714 × 10^−5^ log [GLY] + 0.966 × 10^−5^; R² = 0.983) (Figure 4d). While the resistance to load transfer decreases and it varies linearly as a function of the speed constant k_2_° (k_2_° = −1.563 R_ct2_ + 7.334; R² = 0.993) (Figure 4b) and as a function of the logarithm of the GLY concentration (R_ct2_ = −3547.1 log [GLY] +41865.1; R² = 0.989) (Figure 4c). These variations are recorded with the change from a zero incubated concentration to a concentration of 50 ng/mL of glyphosate. As for the factor n_2_, its value is estimated at 0.96 indicating that the surface is more relatively homogeneous.

Note that the constituent phases have different values of conductivity. R_ct1_, k_1_°, n_1_ and Q_1_ describe the processes that occur in the membrane of chitosan. Consequently, all the variations recorded on these parameters are due to the variation in the concentration of glyphosate in the membrane of MIPs, the greater this concentration; the more the film is resistive, thus causing a decrease in the faradic and capacitive current.

While R_ct2_, k_2_°, n_2_ and Q_2_ correspond to the film of CMA which separates the membrane MIPs from the surface of the sensor, as it can also represent the polarization of the electrode. The impedance of the polarizable electrode can be represented as a parallel connection of the charge transfer resistor and the double layer capacitor. It was assumed in the literature that such a distribution could result from a microscopic roughness always present on the solid surfaces, as it can be caused by a dispersion of the capacitance of interfacial origin, linked to the adsorption of ions and to in homogeneities surface chemicals. The concentration of the redox couple changes in the pores with an exponential gradient. This problem has been addressed in the literature [50,51]. The accumulation of ions on the electrode surface causes an increase in faradic and capacitive current.

The relative variation of the resistance to charge transfer of the electrode is presented using the following equation |R_ct1_ − R_ct1blank_|/R_ct1blank_ (ΔR/R). The calculations of ΔR/R were carried out with Rct1, which is related to the variation in the concentration of GLY. This parameter is found to be linearly proportional to the logarithmic value of GLY concentrations in the range of 0.31 pg/mL to 50 ng/mL, as shown in Figure 5, with a correlation coefficient of 0.999. The linear regression equation is: ΔR/R = 0.221 log[GLY] − 0.02.

The detection limit (LOD) of the considered sensor is estimated at 1 fg/mL. LOD was obtained from the equation 3 S/m [52], where S is the residual standard deviation of the linear regression and m is the slope of the regression line. The results showed a low detection limit and high sensitivity.

The parameters of the CS-MIPs/CMA/Au sensor are compared to another impedance-meter sensor for the detection of GLY which is based on electropolymerized polypyrrole films, as a conductive layer layer between the gold microelectrode surface and the molecularly imprinted chitosan film (CS-MIPs/PPy/Au) [27]. The impedance spectrum of the PPy-based sensor is formed by a single semicircle. This result allows concluding that in the present work, the formation of two half-circles in the impedance spectrum is due to the presence of the CMA. The electron transfer rate in the MIPs films of the CS-MIPs/PPy/Au sensor is much higher (55 µm/s) than that of the CS-MIPs/CMA/Au (5.5 µm/s) sensor due to the presence of the PPy conductive layer which facilitates the electron transfer. The interest of the covalent attachment of the CS-MIPs film on an underlayer is to eliminate the inhomogeneity of the CS-MIPs film caused by the trapping of the H_2_ molecules released during the electrodeposition of the CS film. Due to the insulating property of the CMA film, accumulation of charged species occurred (second semi-circle in the low frequency range), which is not the case using polypyrrole film [27]. 

Despite this enormous difference, the sensitivities of both sensors (with PPy and with CMA), calculated with the relative resistance variation (R/R), are close (0.30 for CS-MIPs/PPy/Au and 0.221 for CS-MIPs/CMA/Au), the detection limits are the same: 1 fg/mL and their possible recyclability is the same. 

### 4.3. Comparison between MIPs and NIP

The response of the NIP sensor was modeled with the model developed. Figure 6 shows the theoretical and experimental evolution of Re (Z) as a function of −Im (Z). A good fit is achieved between the experimental data and the theoretical data with a fairly low error (<10^−3^). The parameters of the NIP sensor are listed in Table 3. These parameters vary insignificantly depending on the concentration of GLY, in particular R_ct1_. From this point, it comes that the non-specific adsorption of glyphosate is very low. 

The coating thickness of the NIP sensor (56.8 µm) is greater than that of the MIPs (36 µm). Therefore, the resistance of the resulting solution is greater on the NIP sensor (238.5 Ohm) compared to MIP (151.14 Ohm).

The value n1 of the MIPs film (n_1_ = 0.79) is lower than that of the NIP (n_1_ = 0.85) which makes it possible to say that the MIPs films are more porous due to the presence of free sites, specific to GLY. This point can also explain the compared values of the electron transfer rate constants: the electron transfer rate constant in the MIPs film is greater (k_1_° = 5.5 µm/s) compared to the NIP film (k_1_° = 1.4 µm/s). The values of n2 characterizing the CMA films are close indicating that there is not a big difference in the morphology of the two films (n_2_ (MIPs) = 0.96 − n_2_ (NIP) = 0.98).

### 4.4. Effects of Differents Parameters on the Impedance Response of CS-MIPs/CMA/Au

#### 4.4.1. Effect of Coefficient *n*

The coefficient n_1_ has an effect on the impedance of the first semicircle which is found at high frequencies. The more n_1_ decreases the more the maximum of –Im (Z) of the first semicircle and its value at the end towards the middle frequencies decreases, in contrast with the minimum value of Re (Z) which increases and its maximum remains constant. The coefficient n_1_ has no effect on the second semi-circle, except for the initial value of the imaginary impedance −Im (Z) (Figure 7a).

With the reduction of the coefficient n_2_ we notice that the maximum of Re.

Experimentally, n_1_ and n_2_ can be modified by varying the thickness of the CMA and MIP films, respectively. These parameters can be controlled by varying the number of cycles during the electrodeposition. In addition, n_2_ can be decreased by the use of a porogene solvent in the synthesis of the CS-MIPs [53], which can cause increase the porosity and tortuosity on the CS film.

(Z) and of –Im (Z) of the first semicircle increases, and that a better definition of the semicircle is obtained for a maximum value n2. For the second semicircle, the maxima of Re (Z) and –Im (Z) in the low frequency domain decrease with the decrease in n_2_. The larger n_2_, the more well defined this semicircle (Figure 7b).

#### 4.4.2. Effect of the Standard Rate Constant k°

With the decrease in the value of k_1_° from 10^−2^ to 10^−3^ cm/s we notice that the real and imaginary impedance of the first semi-circle has increased, for the second semi-circle the maximum of –Im (Z) remained constant and the real impedance increased. The passage at a speed k_1_° of 10^−4^ cm/s recorded a significant increase in the imaginary impedance of the first semicircle, while the maximum of the real impedance remained the same as for k_1_° equal to 10^−3^ cm/s. For the second semicircle, while it is less defined, the maximum value of –Im (Z) is larger and that of Re (Z) is smaller than the values recorded for k_1_° equal to 10^−3^ cm/s. With the decrease in k_1_°, we notice the appearance of a single semicircle with greater resistance (Figure 8), which is observed for the NIP film.

In Figure 9, we have shown the effect of varying the rate constant k_2_° on the impedance spectrum. For large values of k_2_° we notice the appearance of a spectrum with a single semi-circle. With the decrease in k_2_°, two semicircles appear on the spectrum. The maximum of Re (Z) of the first semicircle remains constant, and the maximum of –Im (Z) decreases with the decrease of k_2_°. Whereas, for the second semicircle, the maximum of –Im (Z) decreases and that of Re (Z) increases with the decrease k_2_°. For sufficiently low values of k_2_°, we notice that the semi-circle at low frequencies approaches the shape of a semi-infinite line. 

For obtaining a higher value of k_1_° and k_2_°, the thickness of the two films should be decreased allowing to reduce the charge transfer resistance. On another side, increasing of the concentration of the redox couple should be increased the values of k_1_° and k_2_°. Or, for example, the integration of conductive nanomaterials in the CS and CMA films could also increase the value of k°.

#### 4.4.3. Effect of Coefficient Q

For larger values of Q_1_, the model gives a spectrum with a single semicircle (Figure 10a). With the decrease in the value of Q_1_, we obtain spectra with two semi-circles. The latter is more defined for a minimum Q_1_ value. On the two semicircles, the maximum of −Im (Z) varies little, and the maximum of Re (Z) remains constant.

With the variation of the coefficient of CPE (Q_2_), we obtain a single semicircle for low values of Q_2_. With the increase of Q_2_ the model gives two increasingly better-defined semicircles. The maximum of –Im (Z) varies little, and the maximum of Re (Z) remains constant. For sufficiently large values in Q_2_ we notice the disappearance of part of the second semi-circle which is located towards the low frequency areas (Figure 10b).

So, the sensor is better if the thickness of the electronic double layer is low. Q_1_ and Q_2_ can be decreased by the variation of the movement of the ions by agitation of the electrolyte, by increasing of the temperature, by the variation of the applied potential, or also the changes in the morphology of the CS and CMA films [54,55,56].

## 5. Conclusions

The desire to understand the intrinsic operating mechanisms of the developed microsensor, or to open up new avenues in the implementation of new design processes, requires in-depth knowledge of all the parameters of influence. In this study, the impedimetric response of the CS-MIPs/CMA/Au microsensors has been modeled using mathematical model based on physical laws. The phenomena at the electrode/electrolyte interface were described. The various parameters were determined by minimizing the error between the experimental and theoretical values. Then, the effect of each parameter and GLY concentration on the response signal was shown. The first half-circle in the high frequency range is due to the interaction of glyphosate molecules with the imprinted sites of the CS-MIPs film. The relative variation of the charge transfer resistance is proportional to the log of the concentration of glyphosate. The capacitance decreases as the concentration of glyphosate increases, which is explained by the discharging of the charged imprinted sites when the glyphosate molecule interacts with the imprinted sites through electrostatic interactions. The second half-circle of impedance in the low frequency range is due to the phenomenon of adsorption of the ions in the CMA film, this phenomenon being balanced by the electrostatic interaction of glyphosate with the imprinted sites in the CS-MIPs film. This phenomenon is due to the insulating properties of the CMA film but is not observed when CS-MIPs film is attached to a conductive film such as polypyrrole. Despite these differences of electrochemical behavior, CS-MIPs/CMA/Au and CS-MIPs/PPy/Au microsensors present the same detection limit: 1 fg/mL.

The developed model here for a MIP-chitosan based sensor is applicable to other MIP based sensors. It appears here that electrostatic interactions between template and imprinted sites induce changes in capacitance. The effect of the electrical properties of an underlayer can change the morphology the Nyquist plot. The morphology of the Nyquist plot of the NIP based sensor is totally different from that of the corresponding MIP based sensor.

In order to increase the sensitivity of detection, some improvements could be brought in the preparation of the MIP-based sensor: the thickness of MIP film and of its underlayer should be obtained thinner, the conductivity of the underlayer should be improved by the addition of conductive nanomaterials, the porosity of the MIP film should be improved by using a porogene, the concentration of the redox probe could be increased.

## Figures and Tables

**Figure 1 molecules-27-00493-f001:**
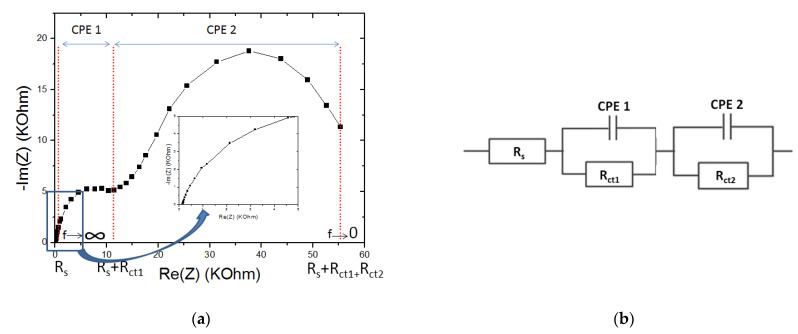
(**a**) EIS of the microsensor (E = 0.2 V. Freq =100 kHz−100 mHz) in PBS with 5 mM [Fe(CN)6]^3−/4−^. (**b**) Equivalent electrical circuit (Rs + CPE1/Rct1+ CPE2/Rct2).

**Figure 2 molecules-27-00493-f002:**
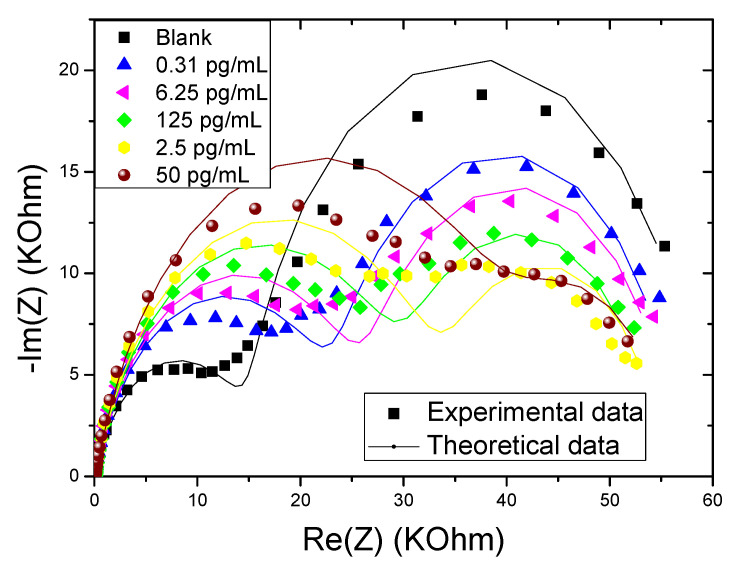
Experimental and theoretical EIS of CS-MIPs/CMA/Au in 5 mM [Fe(CN)6]^3−/4−^ initial potential E = 0.2 V; f = 100 kHz–100 mHz, for different concentrations of GLY (0.31 pg/mL–6.25 pg/mL–125 pg/mL–2.5 ng/mL and 50 ng/mL).

**Figure 3 molecules-27-00493-f003:**
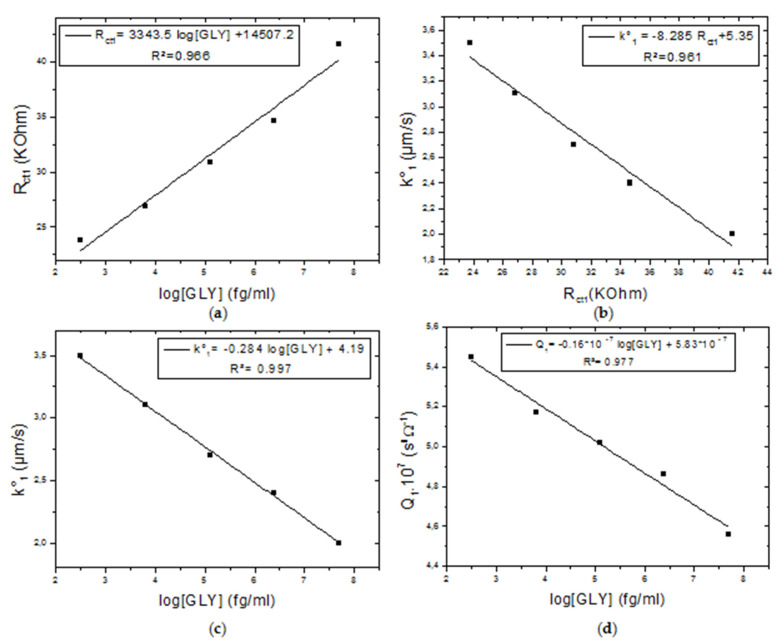
Calibration curves of CS-MIPs/CMA/Au. (**a**) k_1_° = f(log[GLY]), (**b**) k_1_° = f(Rct1) for different [GLY], (**c**) R_ct1_ = f(log[GLY]), (**d**) Q_1_= f(log[GLY]). ([GLY] = 0.31 pg/mL–6.25 pg/mL–125 pg/mL–2.5 ng/mL–50 ng/mL).

**Figure 4 molecules-27-00493-f004:**
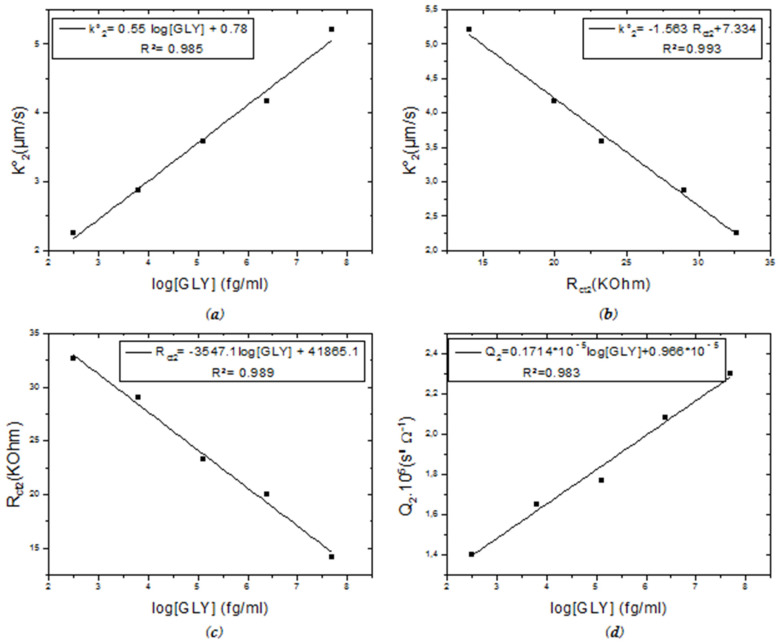
Calibration curves of CS-MIPs/CMA/Au. (**a**) k_2_° = f(log[GLY]), (**b**) k_2_° = f(Rct2) for different [GLY], (**c**) R_ct2_ = f(log[GLY]), (**d**) Q_2_ = f(log[GLY]). ([GLY] = 0.31 pg/mL–6.25 pg/mL–125 pg/mL–2.5 ng/mL–50 ng/mL).

**Figure 5 molecules-27-00493-f005:**
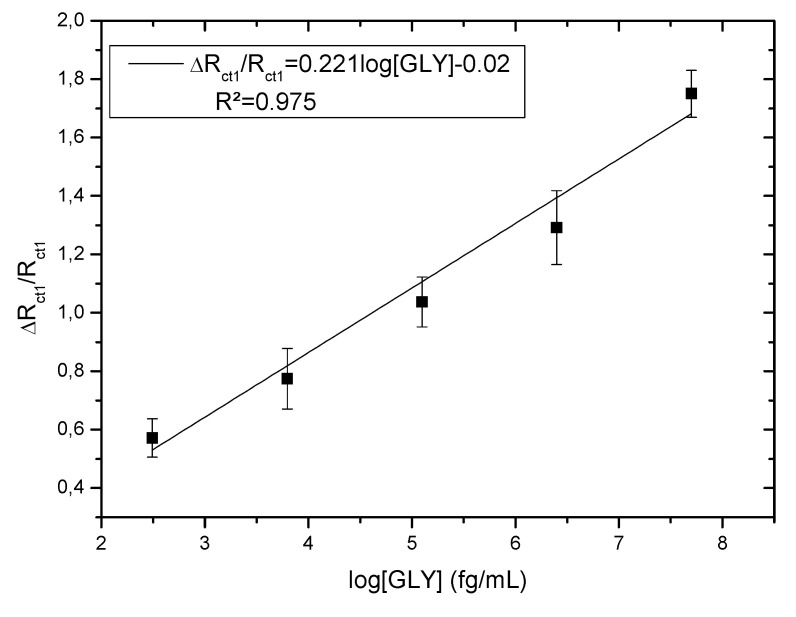
Calibration curves of CS-MIPs/CMA/Au (ΔR/R = 0.221 log[GLY] − 0.02; R² = 0.975).

**Figure 6 molecules-27-00493-f006:**
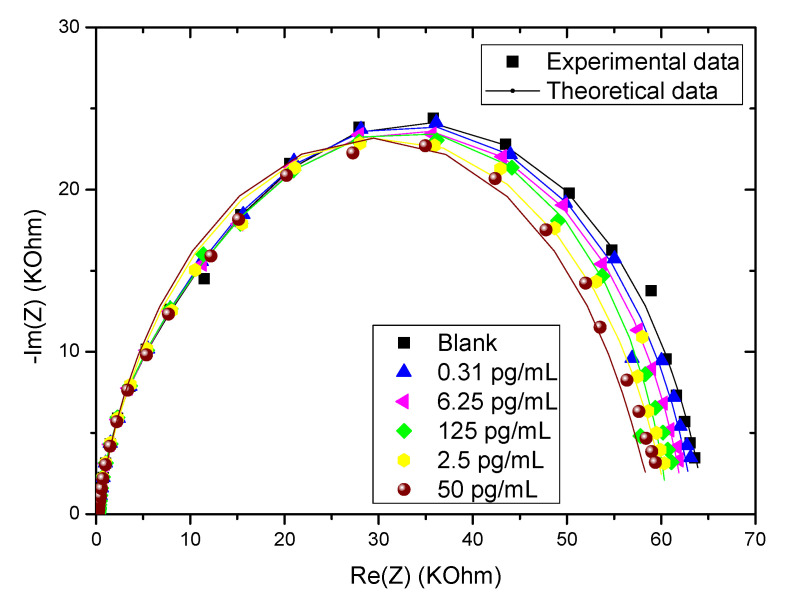
Experimental and theoretical EIS of CS-MIPs/CMA/Au in 5 mM [Fe(CN)6]^3−/4−^ initial potential E = 0.2 V; f = 100 kHz–100 mHz, for different concentrations of GLY (0.31 pg/mL–6.25 pg/mL–125 pg/mL–2.5 ng/mL and 50 ng/mL).

**Figure 7 molecules-27-00493-f007:**
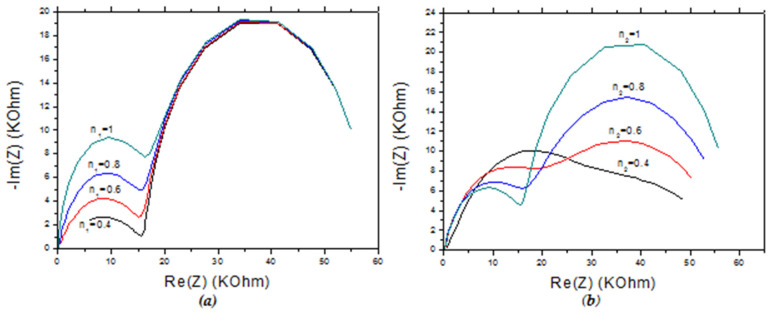
(**a**) Effect of coefficient n1 on the variation of the impedance response of CS-MIPs/CMA/Au for *n*_2_ = 0.95. (**b**) Effect of coefficient n_2_ on the variation of the impedance response of CS-MIPs/CMA/Au for n_1_ =0.8. In 5 mM [Fe(CN)_6_]^3−/4−^. f = 100 kHz–100 mHz. A = 0.0064 cm², T = 298 K, l = 36 µm, k_1_° = 5 µm/s, Q_1_ = 5 × 10^−7^ s^n^ Ω^−1^, k_2_° = 2 µm/s, Q_2_ = 10^−5^ s^n^ Ω^−1^.

**Figure 8 molecules-27-00493-f008:**
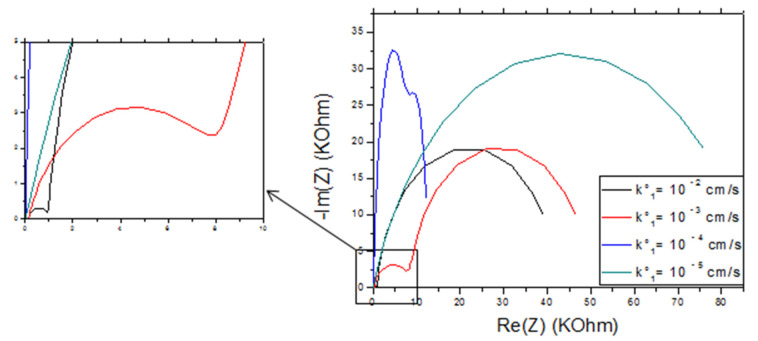
Effect of the rate constant k_1_° on the impedance response of CS-MIPs/CMA/Au. For T = 298 K, A = 0.0064 cm², l = 60µm, Q_1_ = 5 × 10^−7^ s^n^ Ω^−1^, n_1_ = 0.8, k_2_° = 2µm/s, Q_2_ = 10^−5^ s^n^ Ω^−1^, n_2_ = 0.95. [Fe(CN)_6_]^3−/4−^ = 5 mM. f = 100 KHz–100 mHz.

**Figure 9 molecules-27-00493-f009:**
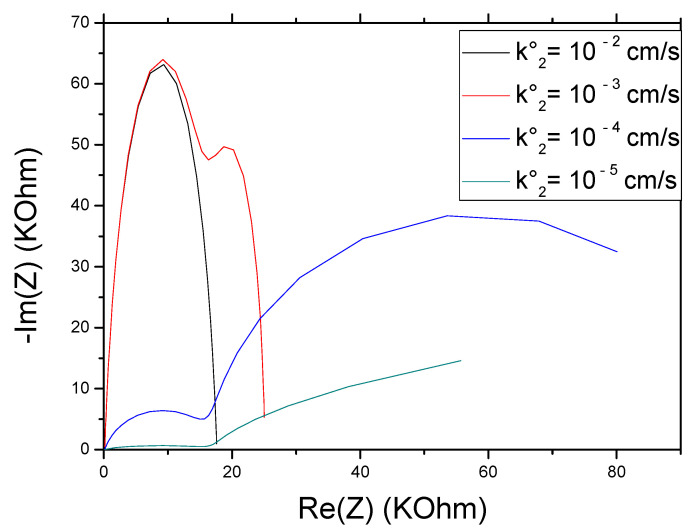
Effect of the rate constant k_2_° on the impedance response of CS-MIPs/CMA/Au. For T = 298 K, A = 0.0064 cm², l = 60µm, k_1_° = 5µm/s, Q_1_ = 5 × 10^−7^, n_1_ = 0.8 s^n^ Ω^−1^, Q_2_ = 10^−5^ s^n^ Ω^−1^, n_2_ = 0.95. [Fe(CN)_6_]^3−/4−^ = 5 mM. f = 100 KHz–100 mHz.

**Figure 10 molecules-27-00493-f010:**
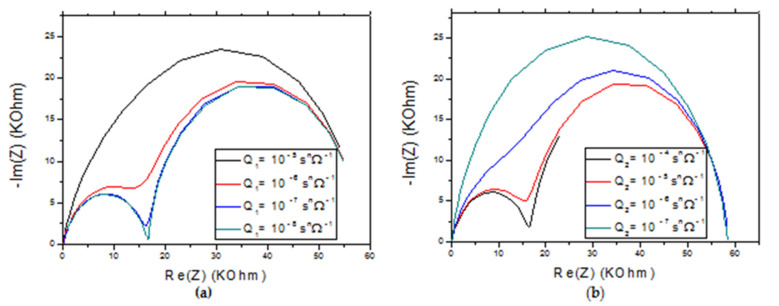
(**a**) Effect of the rate constant Q_1_ on the impedance response of CS-MIPs/CMA/Au for Q_2_ = 10^−5^ s^n^ Ω^−1^. (**b**) Effect of the rate constant Q_2_ on the impedance response of CS-MIPs/CMA/Au for Q_1_ = 5 × 10^−7^. T = 298 K, A = 0.0064 cm², l = 60µm, k_1_° = 5µm/s, Q_1_ = 5 × 10^−7^ s^n^ Ω^−1^, Q_2_ = 10^−5^ s^n^ Ω^−1^, n_1_ = 0.8, n_2_= 0.95. [Fe(CN)_6_]^3−/4−^ = 5 mM. f = 100 KHz–100 mHz.

**Table 1 molecules-27-00493-t001:** Variation interval of different parameters.

Parameters	Variation Range	Unit
*l*	[10^−3^,10^−5^]	cm
k_1_°	[10^−3^,10^−5^]	cm·s^−1^
Q_1_	[10^−5^,10^−8^]	s^n^·Ω^−1^
n_1_	[−1,1]	/
k_2_°	[10^−3^,10^−5^]	cm·s^−1^
Q_2_	[10^−5^,10^−8^]	s^n^·Ω^−1^
n_2_	[−1,1]	/

**Table 2 molecules-27-00493-t002:** Different parameters of CS-MIPs/CMA/Au estimated by the model for different concentrations of GLY.

[GLY]	l (μm)	R_s_ (Ω)	k_1_° (μm/s)	R_ct1_ (Ω)	Q_1_·10^7^ (S^n^ Ω^−1^)	n_1_	k_2_° (μm/s)	R_ct2_ (Ω)	Q_2_·10^5^ (S^n^ Ω^−1^)	n_2_	e_1_ × 10^4^	e_2_ × 10^4^
Blank	36	151.14	5.5	15,132	5.60	0.79	1.92	43,348	1.07	0.96	4.20	3.59
0.31 pg/mL	36	151.14	3.5	23,779	5.45	0.79	2.25	32,638	1.40	0.96	7.04	3.88
6.25 pg/mL	36	151.14	3.1	26,848	5.17	0.79	2.87	28,999	1.65	0.96	6.52	6.84
125 pg/mL	36	151.14	2.7	30,825	5.02	0.79	3.58	23,248	1.77	0.96	7.50	9.94
2.5 ng/mL	36	151.14	2.4	34,678	4.86	0.79	4.17	19,959	2.08	0.96	0.98	0.38
50 ng/mL	36	151.14	2.0	41,614	4.56	0.79	5.21	14,083	2.30	0.96	2.88	9.57

**Table 3 molecules-27-00493-t003:** Different parameters of CS-NIP/CMA/Au estimated by the model for different concentrations of GLY.

[GLY]	l (μm)	R_s_ (Ω)	k_1_° (μm/s)	R_ct1_ (Ω)	Q_1_·10^7^ (S^n^ Ω^−1^)	n_1_	k_2_° (μm/s)	R_ct2_ (Ω)	Q_2_·10^7^ (S^n^ Ω^−1^)	n_2_	e_1_ × 10^4^	e_2_ × 10^4^
Blank	56.8	238.5	1.39	59,563	4.92	0.85	17.02	4885	3.56	0.98	2.22	4.96
0.31 pg/mL	56.8	238.5	1.41	58,850	4.65	0.85	18.60	4469	3.81	0.98	4.15	3.58
6.25 pg/mL	56.8	238.5	1.36	61,271	4.71	0.85	13.09	6349	4.02	0.98	3.54	5.51
125 pg/mL	56.8	238.5	1.45	57,234	4.44	0.85	8.98	9257	4.21	0.98	5.63	8.36
2.5 ng/mL	56.8	238.5	1.39	59,571	4.86	0.85	9.28	8953	3.85	0.98	5.45	4.48
50 ng/mL	56.8	238.5	1.43	58,223	4.64	0.85	9.50	8742	3.66	0.98	4.48	4.22

## Data Availability

The data that support the findings of this study are available from the corresponding author upon reasonable request.

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
