# Peer review of "Mathematical Modelling of Glyphosate Molecularly Imprinted Polymer-Based Microsensor with Multiple Phenomena"

_molecules, 2022, doi:10.3390/molecules27020493_

Round 1

Reviewer 1 Report

The paper present a very good study in the didactical approach to modelling impedance responses in a model systems. It offers a good immersion in the theory of this. The main weakness is that the link to the molecularly imprinted polymer (MIP) for glyphosate is highly ignored and advantages of following this approach are not clearly established.

Abstract needs more numerical information  on the final conclusions from the paper

Introduction needs to add more information on literature about glyphosate analysis using the electrochemical techniques suggested for analysis. Also, the present literature developed using molecularly imprinted polymers for glyphosate is needed, as little reference has been made to them: Information on these will be useful to establish a more in-depth analysis in the discussion section. The introduction lingers for a while on well-known theoretical principles but does not sufficiently explore present literature on the topic of glyphosate analysis.

Materials and methods is thoroughly explained with plenty of theoretical support on the physical principles.

The results and discussion greatly miss the potential influence of the molecularly imprinted polymer and the comparison with NIP and bare electrode when the impedance model is produced. How other polymers, different from chitosan (cited in the literature for this compound) would offer changes in the modelling and the chemical repercussions of these changes are missing in the manuscript. Also, there is a lack of practical analytical comparison with the literature and the usefulness of the modelling, as I commented before. The added section in the introduction can be reused at this point. Without this, the paper results in a good demonstration of how impedance spectroscopy can be modelled but the influence of the main two other actors (glyphosate and the polymer) is missing. Without a strong section explaining and offering more information of these, the paper has little usability or practical use. Anybody trying to find the information useful for analytical purposes, without a clear link to analytical parameters and the advantages of the modelling, will greatly ignore this paper and will avoid citing it. This would be a pity as there is plenty of good science and sections that can be of great use if a better link to practical work can be established. 

The conclusion is very general without going into more specific findings. Parts of this section can be summarised in the abstract to clearly link them together. 

As a summary, a very good paper from the impedance perspective but needing a much more chemical approach in the use of the materials used to modify the electrode surfaces  and how these would affect the modelling. This will be done by establishing more chemical explanations using the chitosan model. As commented, without this, the use of the molecular imprinted polymer in this sensor is just merely anecdotal.  The strength of the paper lies on the clear description of the model so other authors can use it for other polymers and compounds, which it is very difficult in its present form.

Author Response

Manuscript ID: molecules-1526876
Type of manuscript: Article
Title: Mathematical modelling of glyphosate molecularly imprinted polymer-based microsensor with multiple phenomena

Responses to Reviewer 1’s comments

The authors thank the reviewer for valuable comments that will improve the quality of the manuscript.

Reviewer #1

The paper present a very good study in the didactical approach to modelling impedance responses in a model systems. It offers a good immersion in the theory of this. The main weakness is that the link to the molecularly imprinted polymer (MIP) for glyphosate is highly ignored and advantages of following this approach are not clearly established.

Abstract needs more numerical information  on the final conclusions from the paper

Introduction needs to add more information on literature about glyphosate analysis using the electrochemical techniques suggested for analysis. Also, the present literature developed using molecularly imprinted polymers for glyphosate is needed, as little reference has been made to them: Information on these will be useful to establish a more in-depth analysis in the discussion section. The introduction lingers for a while on well-known theoretical principles but does not sufficiently explore present literature on the topic of glyphosate analysis.

The introduction was completed with literature about electrochemical sensors for the detection of glyphosate (lines 41-96)

Materials and methods is thoroughly explained with plenty of theoretical support on the physical principles.

The results and discussion greatly miss the potential influence of the molecularly imprinted polymer and the comparison with NIP and bare electrode when the impedance model is produced. How other polymers, different from chitosan (cited in the literature for this compound) would offer changes in the modelling and the chemical repercussions of these changes are missing in the manuscript. Also, there is a lack of practical analytical comparison with the literature and the usefulness of the modelling, as I commented before. The added section in the introduction can be reused at this point. Without this, the paper results in a good demonstration of how impedance spectroscopy can be modelled but the influence of the main two other actors (glyphosate and the polymer) is missing. Without a strong section explaining and offering more information of these, the paper has little usability or practical use. Anybody trying to find the information useful for analytical purposes, without a clear link to analytical parameters and the advantages of the modelling, will greatly ignore this paper and will avoid citing it. This would be a pity as there is plenty of good science and sections that can be of great use if a better link to practical work can be established. 

From the modelling of the first half-circle in the high frequency range, two main parameters were discussed: Q1 and Rct1.

Q1: lines 309-315

The value of Q1 decreases with increasing GLY concentration. This decrease makes it possible to conclude that the capacitance of the CS-MIPs film and the concentration of GLY are inversely proportional. The same behavior has been experimentally observed for the measured capacitance of the MIPs sensor based on N, N-methylenebisacrylamide cryogel which varies inversely proportional with the concentration of glyphosate 20. The phenomenon was explained by the discharging of the charged imprinted sites when the glyphosate molecule interact with the imprinted sites through electrostatic interactions.

Rct1: lines 372-377

The relative variation of the resistance to charge transfer in Cs-MIPs film is presented using the following equation |Rct1-Rct1blank|/Rct1blank (ΔR/R). The calculations of ΔR/R were carried out with Rct1, which is related to the variation in the concentration of GLY. This parameter is found to be linearly proportional to the logarithmic value of GLY concentrations in the range of 0.31 pg/mL to 50 ng/mL, as shown in Figure 5, with a correlation coefficient of 0.999. The linear regression equation is: ΔR/R = 0.221 log[GLY] - 0.02

The impedimetric behavior of the CS-MIPs/CMA/Au is compared to that of CS-MIPs/PPy/Au: lines 382-395

The parameters of the CS-MIPs/CMA/Au sensor are compared to another impedancemetric sensor for the detection of GLY which is based on electropolymerized polypyrrole films, as a conductive layer between the gold microelectrode surface and the molecularly imprinted chitosan film (CS-MIPs/PPy/Au)26. The impedance spectrum of the PPy-based sensor is formed by a single semicircle. This result allows us to conclude that in the present work, the formation of two half-circles in the impedance spectrum is due to the presence of the CMA. The electron transfer rate in the MIPs films of the CS-MIPs/PPy/Au sensor is much higher (55 µm/s) than that of the CS-MIPs/CMA/Au (5.5 µm/s) sensor due to the presence of the PPy conductive layer which facilitates the electron transfer. The interest of the covalent attachement of the CS-MIPs film on an underlayer is to eliminate the inhomogeneity of the CS-MIPs film caused by the trapping of the H2 molecules released during the electrodeposition of the CS film. Due to the insulating property of the CMA film, accumulation of charged species occured (second semi-circle in the low frequency range), which is not the case using polypyrrole film26.

The conclusion is very general without going into more specific findings. Parts of this section can be summarised in the abstract to clearly link them together. 

The main results are summarized in the conclusion

The desire to understand the intrinsic operating mechanisms of the developed microsensor, or to open up new avenues in the implementation of new design processes, requires in-depth knowledge of all the parameters of influence. In this study, the impedimetric response of the CS-MIPs/CMA/Au microsensors has been modeled using mathematical model based on physical laws. The phenomena at the electrode/electrolyte interface were described. The various parameters were determined by minimizing the error between the experimental and theoretical values. Then, the effect of each parameter and GLY concentration on the response signal was showed. The first half-circle in the high frequency range is due to the interaction of glyphosate molecules with the imprinted sites of the CS-MIPs film. The relative variation of the charge transfer resistance is proportional to the log of the concentration of glyphosate. The capacitance decreases as the concentration of glyphosate increases, which is explained by the discharging of the charged imprinted sites when the glyphosate molecule interact with the imprinted sites through electrostatic interactions. The second half-circle of impedance in the low frequency range is due to the phenomenon of adsorption of the ions in the CMA film, this phenomenon being balanced by the electrostatic interaction of glyphosate with the imprinted sites in the CS-MIPs film. This phenomenon is due to the insulating properties of the CMA film, but is not observed when CS-MIPs film is attached to a conductive film such as polypyrrole. Despite these differences of electrochemical behavior, CS-MIPs/CMA/Au and CS-MIPs/PPy/Au microsensors present the same detection limit: 1 fg/mL.

As a summary, a very good paper from the impedance perspective but needing a much more chemical approach in the use of the materials used to modify the electrode surfaces  and how these would affect the modelling. This will be done by establishing more chemical explanations using the chitosan model. As commented, without this, the use of the molecular imprinted polymer in this sensor is just merely anecdotal.  The strength of the paper lies on the clear description of the model so other authors can use it for other polymers and compounds, which it is very difficult in its present form.

Reviewer 2 Report

Review: molecules-1526876.

Title: Mathematical modelling of glyphosate molecularly imprinted polymer-based microsensor with multiple phenomena.

In this manuscript Authors described a mathematical model of molecularly imprinted polymer (MIP) for glyphosate detection by electrochemical sensor. The model was correlated to experimental data, allowing to predict various parameters of the impedance response of the sensor. In my opinion, the mathematical model approach to screen potential variables for molecularly imprinted sensor behavior is interesting. However, a few important concerns could be identified at this stage of the manuscript evaluation:

  1. The universal aspects of the model should be discussed in detail. One could expect that the presented model could be only applicable to glyphosate imprinted polymer since MIPs are characterized by high specificity. Thus, Authors shall prove if the model could be used for general purpose to other MIP sensor systems. Otherwise, Authors shall justify convincingly the necessity to elaborate the specific model. On the other hand, chitosan-based MIP system could be characterized by high non-specific adsorption (but reference data is omitted).
  2. Taking into account previous paper of Authors (see: Front. Chem. 2021, 9, 621057) very clear distinction between the results published in above mentioned paper and new results have to be indicated with proper reference (for instance: Fig. 5, in Front. Chem. 2021, 9, 621057 and Fig. 2, in a reviewed manuscript). Please, refer also to other papers (see: Sens. Actuators B 2020, 309, 127753) and discuss the model with the context of published results. Please, specify clearly experimental data that was transferred from published sources (line 214).
  3. Introduction suffers from insufficient discussion of novelty with respect to recent papers devoted to mathematic models of MIP sensors (see: Chemosensors 2020, 8 104), other model systems (see: Microchim. Acta 2017, 184, 1959, ACS Omega 2021, 6, 27007, Sensors 2021, 21, 296) as well as reference to recent MIP reviews emphasizing the advantages of those materials (see: Trends Anal. Chem. 2020, 130, 115982).
  4. Conclusions should be rewritten. This section should emphasize the findings presented in the manuscript and should refer to main achievements of this study. For instance, a conclusion provided in line 390. By the way, the mean thickness of the layer of chitosan on functionalized 4-aminophenylacetic acid was equal to 36 µm – please explain if the thickness was constant in batch-to-batch experiments?
  5. The editorial and typographic errors shall be corrected. Captions of figures – superscripts and subscripts are omitted almost everywhere as well as in the text (i.e. line 248), refs. 30-32 (line 392) are missing. Sentence (lines 241-242) shall be checked for correctness. Please define term ‘certain aspects’ (line 74). Parentheses shall be closed in line 58.

I hope that above mentioned suggestions  will strengthen the scientific value of the manuscript. In my opinion major revision is required.

Author Response

Manuscript ID: molecules-1526876
Type of manuscript: Article
Title: Mathematical modelling of glyphosate molecularly imprinted polymer-based microsensor with multiple phenomena

Responses to Reviewer 2’s comments

The authors thank the reviewer for valuable comments that will improve the quality of the manuscript.

Reviewer #2

In this manuscript Authors described a mathematical model of molecularly imprinted polymer (MIP) for glyphosate detection by electrochemical sensor. The model was correlated to experimental data, allowing to predict various parameters of the impedance response of the sensor. In my opinion, the mathematical model approach to screen potential variables for molecularly imprinted sensor behavior is interesting. However, a few important concerns could be identified at this stage of the manuscript evaluation:

  1. The universal aspects of the model should be discussed in detail. One could expect that the presented model could be only applicable to glyphosate imprinted polymer since MIPs are characterized by high specificity. Thus, Authors shall prove if the model could be used for general purpose to other MIP sensor systems. Otherwise, Authors shall justify convincingly the necessity to elaborate the specific model. On the other hand, chitosan-based MIP system could be characterized by high non-specific adsorption (but reference data is omitted).

The application of the developed model for other type of MIP based sensors is discussed in the conclusion part

The developed model here for a MIP-chitosan based sensor is applicable to other MIP based sensors. It appears here that electrostatic interactions between template and imprinted sites induce changes in capacitance. The effect of the electrical properties of an underlayer can change the morphology the Nyquist plot. The morphology of the Nyquist plot of the NIP based sensor is totally different from that of the corresponding MIP based sensor.

  1. Taking into account previous paper of Authors (see: Front. Chem. 2021, 9, 621057) very clear distinction between the results published in above mentioned paper and new results have to be indicated with proper reference (for instance: Fig. 5, in Front. Chem. 2021, 9, 621057 and Fig. 2, in a reviewed manuscript). Please, refer also to other papers (see: Sens. Actuators B 2020, 309, 127753) and discuss the model with the context of published results. Please, specify clearly experimental data that was transferred from published sources (line 214).

All previous references were cited in the introduction

The choice of the functional monomer in the synthesis of MIPs is an essential step given its capacity to provide complementary interactions with the target molecules 21. Chitosan (CS) is a natural polysaccharide obtained by deacetylation of chitin which is the most abundant non-toxic, biodegradable and biocompatible natural amino polysaccharide 22. Chitosan has three types of reactive functional groups, an amine group and two primary and secondary hydroxyl groups. The presence of reactive functional groups on the polysaccharide chain of CS gives it flexibility, excellent film-forming capacity, good adhesion, biocompatibility, high mechanical resistance, possibility of undergoing structural modifications, which makes it suitable for the preparation of MIPs and for the construction of electrochemical sensors 23,24.

CS-MIPs electrochemical sensors are based on the electrodeposition of CS on the surface of the electrodes 25,26. The electrodeposition of chitosan on a conductive surface by pH gradient causes release of H2. The trapping of these molecules inside the film makes it non-homogeneous 27. The grafting of chitosan on the diazonium salt by the formation of strong interactions probably leads to more homogeneous membranes and to the resolution of interface problems 28.

Experimental results transferred from published sources are cited (lines 279-280)

  1. Introduction suffers from insufficient discussion of novelty with respect to recent papers devoted to mathematic models of MIP sensors (see: Chemosensors 2020, 8 104), other model systems (see: Microchim. Acta 2017, 184, 1959, ACS Omega 2021, 6, 27007, Sensors 2021, 21, 296) as well as reference to recent MIP reviews emphasizing the advantages of those materials (see: Trends Anal. Chem. 2020, 130, 115982).

The MIP-based electrochemical sensors for the detection of glyphosate were cited and discussed in the introduction

Due to the need to overcome these limitations, an alternative has been developed which consists of using molecularly imprinted polymers (MIPs). MIPs were used as a specific recognition material having binding sites of size and shape complementary to target molecules (template). To generate cavity-selective MIPs, a monomer is simply polymerized in the presence of the target molecule and a crosslinking agent resulting in the formation of a highly crosslinked template/monomer complex. 13,14. MIPs are characterized by very interesting properties such as physical resistance, robustness, resistance to high pressures and temperatures and high inertia towards various chemicals 15. In addition, MIPs have high affinity and selectivity towards target molecules; they are comparable to natural receptors 16. Sensors based on molecularly imprinted polymers (MIPs) are being developed for the detection of GLY. The functional monomers most often used in synthesis are MCA (N-methacryloyl-L-cysteine) 17and PPy (Polypyrrole) 1819. In these works, the electrochemical detection of glyphosate was obtained by voltammetric methods. A more recent studiy, using MIP based on polyacrylamide cryogels, glyphosate was detected through capacitive measurements 20.

The choice of the functional monomer in the synthesis of MIPs is an essential step given its capacity to provide complementary interactions with the target molecules 21. Chitosan (CS) is a natural polysaccharide obtained by deacetylation of chitin which is the most abundant non-toxic, biodegradable and biocompatible natural amino polysaccharide 22. Chitosan has three types of reactive functional groups, an amine group and two primary and secondary hydroxyl groups. The presence of reactive functional groups on the polysaccharide chain of CS gives it flexibility, excellent film-forming capacity, good adhesion, biocompatibility, high mechanical resistance, possibility of undergoing structural modifications, which makes it suitable for the preparation of MIPs and for the construction of electrochemical sensors 23,24.

CS-MIPs electrochemical sensors are based on the electrodeposition of CS on the surface of the electrodes 25,26. The electrodeposition of chitosan on a conductive surface by pH gradient causes release of H2. The trapping of these molecules inside the film makes it non-homogeneous 27. The grafting of chitosan on the diazonium salt by the formation of strong interactions probably leads to more homogeneous membranes and to the resolution of interface problems 28.

  1. Conclusions should be rewritten. This section should emphasize the findings presented in the manuscript and should refer to main achievements of this study. For instance, a conclusion provided in line 390. By the way, the mean thickness of the layer of chitosan on functionalized 4-aminophenylacetic acid was equal to 36 µm – please explain if the thickness was constant in batch-to-batch experiments?

Conclusions were rewritten according to the main results and possible improvements:

Then, the effect of each parameter and GLY concentration on the response signal was shown. The first half-circle in the high frequency range is due to the interaction of glyphosate molecules with the imprinted sites of the CS-MIPs film. The relative variation of the charge transfer resistance is proportional to the log of the concentration of glyphosate. The capacitance decreases as the concentration of glyphosate increases, which is explained by the discharging of the charged imprinted sites when the glyphosate molecule interact with the imprinted sites through electrostatic interactions. The second half-circle of impedance in the low frequency range is due to the phenomenon of adsorption of the ions in the CMA film, this phenomenon being balanced by the electrostatic interaction of glyphosate with the imprinted sites in the CS-MIPs film. This phenomenon is due to the insulating properties of the CMA film, but is not observed when CS-MIPs film is attached to a conductive film such as polypyrrole. Despite these differences of electrochemical behavior, CS-MIPs/CMA/Au and CS-MIPs/PPy/Au microsensors present the same detection limit: 1 fg/mL.

The developed model here for a MIP-chitosan based sensor is applicable to other MIP based sensors. It appears here that electrostatic interactions between template and imprinted sites induce changes in capacitance. The effect of the electrical properties of an underlayer can change the morphology the Nyquist plot. The morphology of the Nyquist plot of the NIP based sensor is totally different from that of the corresponding MIP based sensor.

In order to increase the sensitivity of detection, some improvements could be brought in the preparation of the MIP-based sensor: the thickness of MIP film and of its underlayer should be obtained thinner, the conductivity of the underlayer should be improved by the addition of conductive nanomaterials, the porosity of the MIP film should be improved by using a porogene, the concentration of the redox probe could be increased.

The mean thickness of the layer of chitosan on functionalized 4-aminophenylacetic acid was equal to 36 µm

This value was found to vary of less than 1 µm from one sensor to another. (cf line 290).

  1. The editorial and typographic errors shall be corrected. Captions of figures – superscripts and subscripts are omitted almost everywhere as well as in the text (i.e. line 248), refs. 30-32 (line 392) are missing. Sentence (lines 241-242) shall be checked for correctness. Please define term ‘certain aspects’ (line 74). Parentheses shall be closed in line 58.

All these points were corrected

I hope that above mentioned suggestions  will strengthen the scientific value of the manuscript. In my opinion major revision is required.

Round 2

Reviewer 2 Report

Review: molecules-1526876_R1.

Title: Mathematical modelling of glyphosate molecularly imprinted polymer-based microsensor with multiple phenomena.

This is a revised version of the manuscript. Authors have made corrections according to referee comments. In my opinion, the manuscript in current form could be considered for acceptance. Only minor suggestion is related to the papers cited by Authors in lines 66-75, refs. 13-16, which in my opinion should be replaced by newest citations from the field of MIP synthesis and application (see: see: Chem. Rev. 2019, 119, 94, Materials 2021, 14, 1850, Chemosensors 2021, 9, 123, Polymers 2020, 12, 1154, Sensors 2019, 19, 1279).

Therefore, minor revision could be suggested before final decision of the Editor.

Author Response

Manuscript: molecules-1526876_R1.

Title: Mathematical modelling of glyphosate molecularly imprinted polymer-based microsensor with multiple phenomena.

Reviewer #2

This is a revised version of the manuscript. Authors have made corrections according to referee comments. In my opinion, the manuscript in current form could be considered for acceptance. Only minor suggestion is related to the papers cited by Authors in lines 66-75, refs. 13-16, which in my opinion should be replaced by newest citations from the field of MIP synthesis and application (see: see: Chem. Rev. 2019, 119, 94, Materials 2021, 14, 1850, Chemosensors 2021, 9, 123, Polymers 2020, 12, 1154, Sensors 2019, 19, 1279).

Therefore, minor revision could be suggested before final decision of the Editor.

The references 13-17 were replaced by the newest references suggested by the reviewer.